

# EURO-SUPREME: sub-daily precipitation extremes in the EURO-CORDEX ensemble

Anouk Dierickx[1,2], Wout Dewettinck[2], Bert Van Schaeybroeck[3,4], Lesley De Cruz[5,6],
Steven Caluwaerts[2,4], Piet Termonia[2,4], and Hans Van de Vyver[4]

[1]Environmental and Applied Fluid Dynamics Department, von Karman Institute for Fluid Dynamics, Sint-Genesius-Rode, Belgium
[2]Department of Physics and Astronomy, Ghent University, Ghent, Belgium
[3]Department of Geography, Ghent University, Ghent, Belgium
[4]Meteorological and Climatological Research, Royal Meteorological Institute, Brussels, Belgium
[5]Department of Observations, Royal Meteorological Institute, Brussels, Belgium
[6]Department of Electronics and Informatics, Vrije Universiteit Brussel, Brussels, Belgium

**Correspondence:** Hans Van de Vyver (hvijver@meteo.be)

**Abstract.** Extreme precipitation events can lead to devastating floods, loss of life and severe infrastructure damage and are expected to increase in a warming world, highlighting the urgent need to quantify current-day and future extremes. Although intense precipitation extremes are generally better represented by high-resolution climate models, large ensemble datasets are lacking. Yet, these are very essential for estimating uncertainties of future trends. Here, the EURO-SUPREME dataset, with

DOI: doi.org/10.26050/WDCC/EURCORDEX_prec (Van de Vyver et al., 2024), is presented that includes extreme precipitation events from the EURO-CORDEX 0.11° ensemble (coupled to CMIP5) for accumulated precipitation depths ranging from 1 hour to 72 hours. More specifically, the data are provided by a 35-member ensemble of historical and future (RCP8.5) annual-maxima on the EURO-CORDEX domain, covering 4984 simulation years in total. The resource is designed to enable climate-model benchmarking and support various state-of-the-art scientific research efforts and climate-change risk assess-

ments. We provide a validation of the EURO-SUPREME dataset for various countries in Europe and investigate the changes in intensity and frequency of extreme precipitation under different global warming levels. Furthermore, we disentangle the RCM and GCM contributions to the biases. Finally, we provide a practical application of using EURO-SUPREME as a benchmark for high-resolution convection-permitting models for Belgium.

## 1 Introduction

Extreme precipitation events are catastrophic by nature and often cause flooding, infrastructure damage and agricultural losses, with significant socio-economic impacts and threats to human security (Tradowsky et al., 2023; Kimutai et al., 2024). In 2023 alone, 1.6 million people in Europe were affected by floods (Copernicus Climate Change Service (C3S), 2024). The sixth assessment report of the Intergovernmental Panel on Climate Change (IPCC, 2023) indicates that both the frequency and intensity of extreme precipitation events have increased since the 1950s in most regions around the world, as shown in

e.g. Westra et al. (2013). This trend is expected to continue as global temperatures continue to rise.



Sub-daily extreme precipitation, mainly caused by convective events, has significant impacts on various sectors, including agriculture and urban water management. In urban areas, for example, heavy rainfall within short periods can lead to flash floods. In recent years, there has been mounting evidence that the intensity of short-term precipitation events is increasing worldwide (Westra et al., 2014; Fowler et al., 2021; Cannon et al., 2024).

It is essential to develop climate models that can adequately represent these events. High-resolution regional climate models (RCMs) are generally considered more accurate for simulating precipitation extremes than global circulation models (GCMs) (Jones et al., 1995; Durman et al., 2001; Doblas-Reyes et al., 2021). The Coordinated Regional Climate Downscaling Experiment (CORDEX) aims to advance and coordinate the science and application of regional climate downscaling and has led to the production of large ensembles of coordinated RCM simulations, including simulations at $0.11°$ (12.5 km) resolution over Europe (Jacob et al., 2014, 2020). However, the ability of climate and weather models with resolution above 5 km to accurately simulate sub-daily precipitation extremes is limited because they rely on parameterisations of deep convection.

Convection-permitting (CPMs) models, which operate at higher resolutions (typically 1-4 km), partly resolve convective processes and produce more realistic short-duration extremes (Prein et al., 2015; Lucas-Picher et al., 2021). However, their required computing power is much larger than that of RCMs, making simulations over a large area exceptional (Kendon et al., 2021) and multi-model ensembles over smaller areas rare (Pichelli et al., 2021). The added value of CPMs for simulating sub-daily precipitation extremes can be confirmed by using coarse-resolution RCMs as a benchmark, but EURO-CORDEX extremes are not readily available because processing the full hourly dataset and extracting the extremes is very time-consuming, technically complicated, computationally demanding and requires a huge storage capacity.

To meet this demand, we introduce the EURO-SUPREME dataset of annual maximum hourly to 72 h rainfall across Europe from a 35-member ensemble of EURO-CORDEX simulations at a spatial resolution of $0.11°$. Similar datasets over the European domain were produced from a large multi-model ensemble of EURO-CORDEX simulations for extreme wind and 5-day precipitation (Outten and Sobolowski, 2024), and for extreme sub-daily precipitation from a single model (CRCM5) initial-condition large ensemble (Poschlod et al., 2021).

The here-presented EURO-SUPREME dataset can be used for several purposes and by different communities. As suggested above, a first important application is to serve as a benchmark set, and in addition to the usual bias analysis, physically-based metrics (such as daily/seasonal cycle, spatial organisation) can also be involved to gain more confidence in the simulations (Cortés-Hernández et al., 2016; Van de Vyver et al., 2021). Comparative studies between the EURO-CORDEX $0.11°$ ensemble and convection-permitting simulations of hourly extreme precipitation can be found in Meredith et al. (2021) for Germany, and in Dierickx (2024) for the UK. Secondly, other applications may involve e.g. inter-model cross-validation of bias-correction techniques (Schmith et al., 2021; Van de Vyver et al., 2023), or attribution of heavy precipitation events (Kimutai et al., 2024). Finally, the dataset can be used for climate risk assessment. Indeed, EURO-CORDEX is included as the reference dataset in the technical screening criteria of the EU Taxonomy Regulations (European Commission, 2022) for undertakings to evaluate climate-related risk (Canepa, 2023).

In Sect. 2 of this work, we describe the EURO-CORDEX ensemble and the EURO-SUPREME dataset structure and its accessibility. In Sect. 3, the data quality is assessed by comparing 10-year return levels of EURO-SUPREME to observations





for different regions over Europe, and the GCM–RCM contributions herein are further explored. Changes in the intensity and frequency of simulated precipitation extremes influenced by global warming are investigated in Sect. 4. In Sect. 5, we discuss the EURO-SUPREME dataset usability by providing potential applications, including benchmarking for CPMs. Finally, in Sect. 8, conclusions are drawn and possible limitations and possibilities for further research are discussed.

## 2 Data

### 2.1 EURO-CORDEX ensemble

EURO-CORDEX is a collaborative initiative aimed at providing coordinated high-resolution climate projections for Europe (Jacob et al., 2014, 2020; Kotlarski et al., 2014), launched as part of the larger CORDEX framework which is a scientific effort of the World Climate Research Program (WCRP). RCM-simulations for the European domain were performed at two
spatial resolutions: the standard CORDEX resolution of $0.44°$ (EUR-44, $\sim 50$ km) and a higher resolution of $0.11°$ (EUR-11, $\sim 12.5$ km). The GCMs that provide the lateral boundary conditions for the RCMs are from the CMIP5 ensemble, with historical simulations for the period 1850-2005 and future projections from 2006 to 2100 using different emission scenarios (e.g. RCP4.5, RCP8.5). CMIP6-CORDEX, which follows SSP scenarios, is currently being implemented and is expected to progress in the coming years (Katragkou et al., 2024).

EURO-CORDEX data are publicly available on Earth System Grid Federation (ESGF) nodes with rainfall at hourly temporal resolution available for 27 GCM-RCM pairs in the $0.11°$ ensemble (see Table 1), and from these we selected the historical and RCP8.5 experiments with the aim of obtaining the largest signal-to-noise ratio. There are in total six different RCMs and six different GCMs. Some of these GCM-RCM pairs are represented by multiple realisations (ensemble members), ultimately resulting in an ensemble of 35 members. Table S1 provides more technical dataset details such as spatial domain and simulation
periods associated with each RCM. Note that there is no standard EUR-11 grid and different RCMs may have different grids.

### 2.2 EURO-SUPREME dataset description

The EURO-SUPREME files include annual-maximum rainfall data and are available in Network Common Data Format (NetCDF) on the World Data Center for Climate (WDCC) via: doi.org/10.26050/WDCC/EURCORDEX_prec (Van de Vyver et al., 2024). More specifically, within each file, we provide for each grid point and for each year, the annual maximum
of $d$-hour accumulated precipitation over a 1-hour moving window on the native RCM grid. The rainfall durations are $d = 1, 2, 3, 4, 6, 8, 12, 24, 48$ and 72 h. The data are organised in a hierarchical structure with layers: Projects, Experiments (with above DOI), and datasets grouped per RCM. Each NetCDF file contains annual maxima of one RCM downscaling of one GCM, and is named using the following convention:

AM_pr_EUR-11_[*GCM name and version*]_[*period*]_[*member*]_[*RCM name and version*].nc





where *period* is either "hist" or "rcp85", and *member* follows the RIP-nomenclature (RIP = realisation, initialisation, physics). The variable names in the NetCDF files are: "am[*d*]h", with rainfall duration *d* equal to "1", "2",..., "72". The spatial dimensions (longitude/latitude) are adapted from the original RCM data. The time dimension represents the year of

the annual maximum. In total, the dataset consists of 70 files, resulting from 35 simulations, each with 2 periods. The dataset contains 4984 years of simulated annual maxima and has a size of 18 GB.

A first analysis revealed extremely high hourly precipitation amounts in the simulations, possibly attributable to the occurrence of non-physical "grid point storms" (Kendon et al., 2023). Such anomalies occur, for instance, in RegCM4-6, where we found rainfall in excess of 8000 mm over North Africa. Hourly values up to 300-350 mm are seen in HadREM3-GA7-05 and

REMO2015 over the Mediterranean Sea, often at the coast. Hourly values of 150-200 mm are also found very sporadically in COSMO-crCLIM and to an even lesser extent in RCA4. However, a strict separation between a grid point storm and a physically-realistic value is not always clear, as extreme hourly values of up to 305 mm have already been observed (WMO, 2024). ALADIN63, on the other hand, did not produce values in excess of 80 mm.

Fig. 1-2 show maps with 10-year return levels of hourly and 24-hour precipitation, respectively, for the EURO-SUPREME

ensemble for the historical period. The return levels were calculated with the annual-maximum method described in Appendix A. It is clear that the 10-year return-level maps for hourly precipitation are very similar per column, i.e. for simulations which use the same RCM but a different forcing GCM. On average, HadREM3-GA7-05 shows higher return levels of hourly precipitation than all other RCMs, especially over the northern part of the Mediterranean. ALADIN63 and RegCM4-6, on the contrary, generally have lower return levels. For daily precipitation extremes, the differences between the RCMs are less clear

(Fig. 2). For all simulations, the highest return levels appear over the Mediterranean coasts. Again, the lowest and highest daily return levels are found for RegCM4-6 and HadREM3-GA7-05, respectively, but their difference is strongly reduced with respect to hourly precipitation.





**Table 1.** Summary of the 27 EURO-CORDEX GCM-RCM pairs that are included in the EURO-SUPREME dataset.

| Institute | RCM | GCM (CMIP5) | GCM member |
|---|---|---|---|
| CNRM (Toulouse, France) | ALADIN63 | CNRM-CERFACS-CNRM-CM5 | r1 |
| | | MOHC-HadGEM2-ES | r1 |
| | | MPI-M-MPI-ESM-LR | r1 |
| | | NCC-NorESM1-M | r1 |
| CLMcom-ETH (ETH Zurich, Switzerland & CLM-Community) | COSMO-crCLIM | CNRM-CERFACS-CNRM-CM5 | r1 |
| | | ICHEC-EC-EARTH | r12, r1, r3 |
| | | MOHC-HadGEM2-ES | r1 |
| | | MPI-M-MPI-ESM-LR | r1, r2, r3 |
| | | NCC-NorESM1-M | r1 |
| MOHC (Exeter, UK) | HadREM3-GA7-05 | CNRM-CERFACS-CNRM-CM5 | r1 |
| | | ICHEC-EC-EARTH | r12 |
| | | MOHC-HadGEM2-ES | r1 |
| | | MPI-M-MPI-ESM-LR | r1 |
| | | NCC-NorESM1-M | r1 |
| SMHI (Norrköping, Sweden) | RCA4 | CNRM-CERFACS-CNRM-CM5 | r1 |
| | | ICHEC-EC-EARTH | r12, r1, r3 |
| | | IPSL-IPSL-CM5A-MR | r1 |
| | | MOHC-HadGEM2-ES | r1 |
| | | MPI-M-MPI-ESM-LR | r1, r2, r3 |
| | | NCC-NorESM1-M | r1 |
| ICTP (Grignano, Italy) | RegCM4-6 | CNRM-CERFACS-CNRM-CM5 | r1 |
| | | ICHEC-EC-EARTH | r12 |
| | | MPI-M-MPI-ESM-LR | r1 |
| | | NCC-NorESM1-M | r1 |
| GERICS (Hamburg, Germany) | REMO2015 | CNRM-CERFACS-CNRM-CM5 | r1 |
| | | IPSL-IPSL-CM5A-MR | r1 |
| | | MPI-M-MPI-ESM-LR | r3 |

## 3 Validation

We validate the EURO-SUPREME dataset by comparing 10-year return levels from historical simulations with observed return levels. Due to the limited availability of hourly observation data, we used national extreme-value statistics from six European countries that were also already included in the validation of Berg et al. (2019) and Poschlod and Ludwig (2021). More specifically, gridded observational return levels are available for Belgium (Van de Vyver, 2012, 2013) and Germany (DWD Climate Data Center (CDC), 2020; Malitz and Ertel, 2015; Junghänel et al., 2017). Even though no gridded observational return levels are available for the Netherlands, Denmark and Finland, country-averaged return levels for each duration exist for these countries (Table S2). Finally, for Britain (UK), return levels were calculated based on CEH-GEAR1hr (Lewis et al., 2018, 2022), a gridded dataset of observed hourly precipitation with a resolution of 1 km.

Since RCM simulations represent areal rather than point precipitation (with an area of 0.11° x 0.11°) and are also stored with hourly resolution, a fair comparison with station-based observed return levels is established by dividing them by the temporal-areal reduction factors of Berg et al. (2019, Table 2), also given in our Table S3.

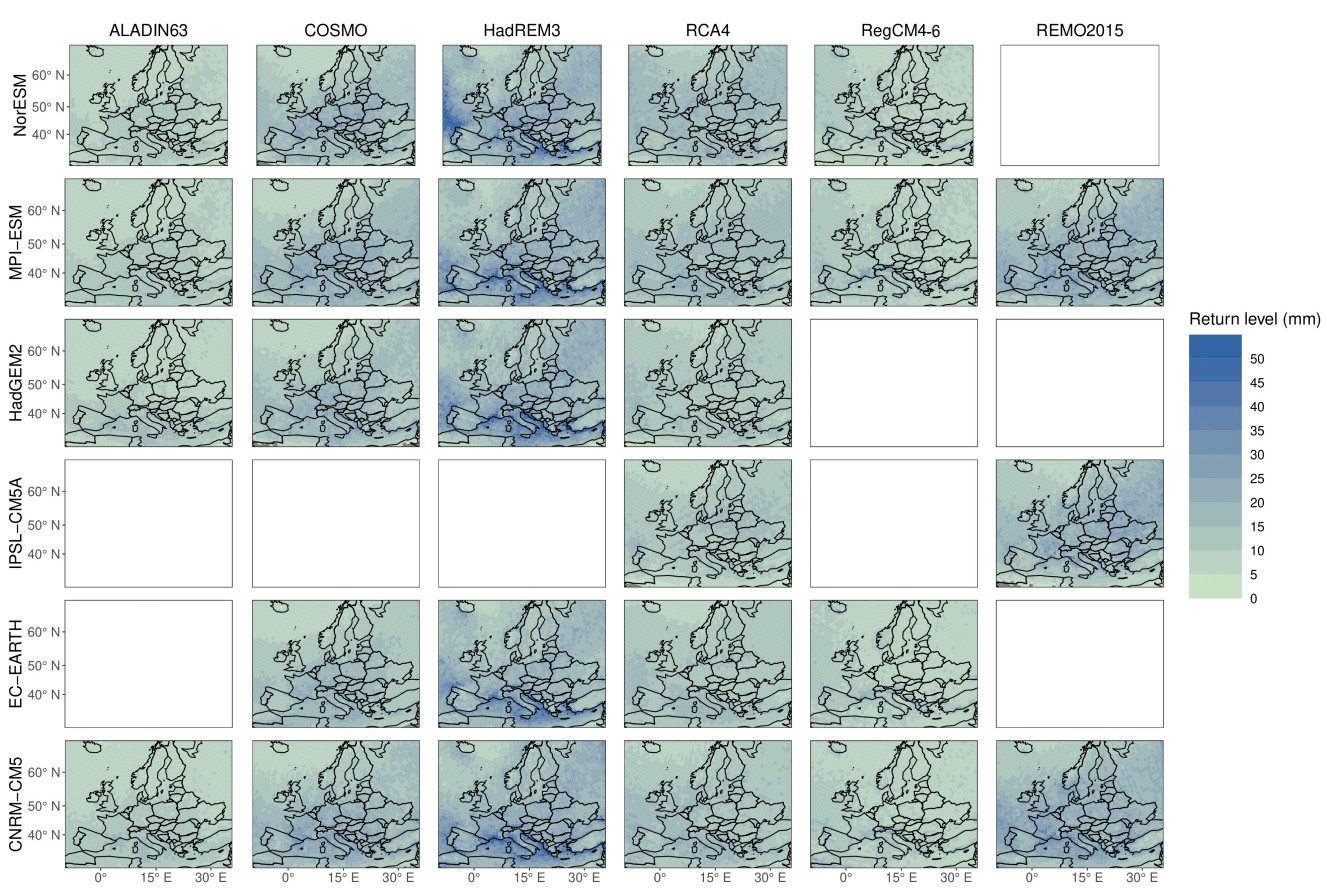

**Figure 1.** 10-year return level maps of hourly precipitation in the historical EURO-SUPREME dataset for the whole domain. Each row corresponds to a GCM and each column to an RCM. The GCM names are shortened: CNRM-CM5 stands for CNRM-CERFACS-CNRM-CM5, EC-EARTH for ICHEC-EC-EARTH, IPSL-CM5A for IPSL-IPSL-CM5A-MR, HadGEM2 for MOHC-HadGEM2-ES, MPI-ESM for MPI-M-MPI-ESM-LR and NorESM for NCC-NorESM1-M. Some of the RCM names are shortened as well: COSMO stands for COSMO-crCLIM and HadREM3 for HadREM3-GA7-05. The darkest colour shade is used for all values above 50 mm and the maximum value found is 77.8 mm.

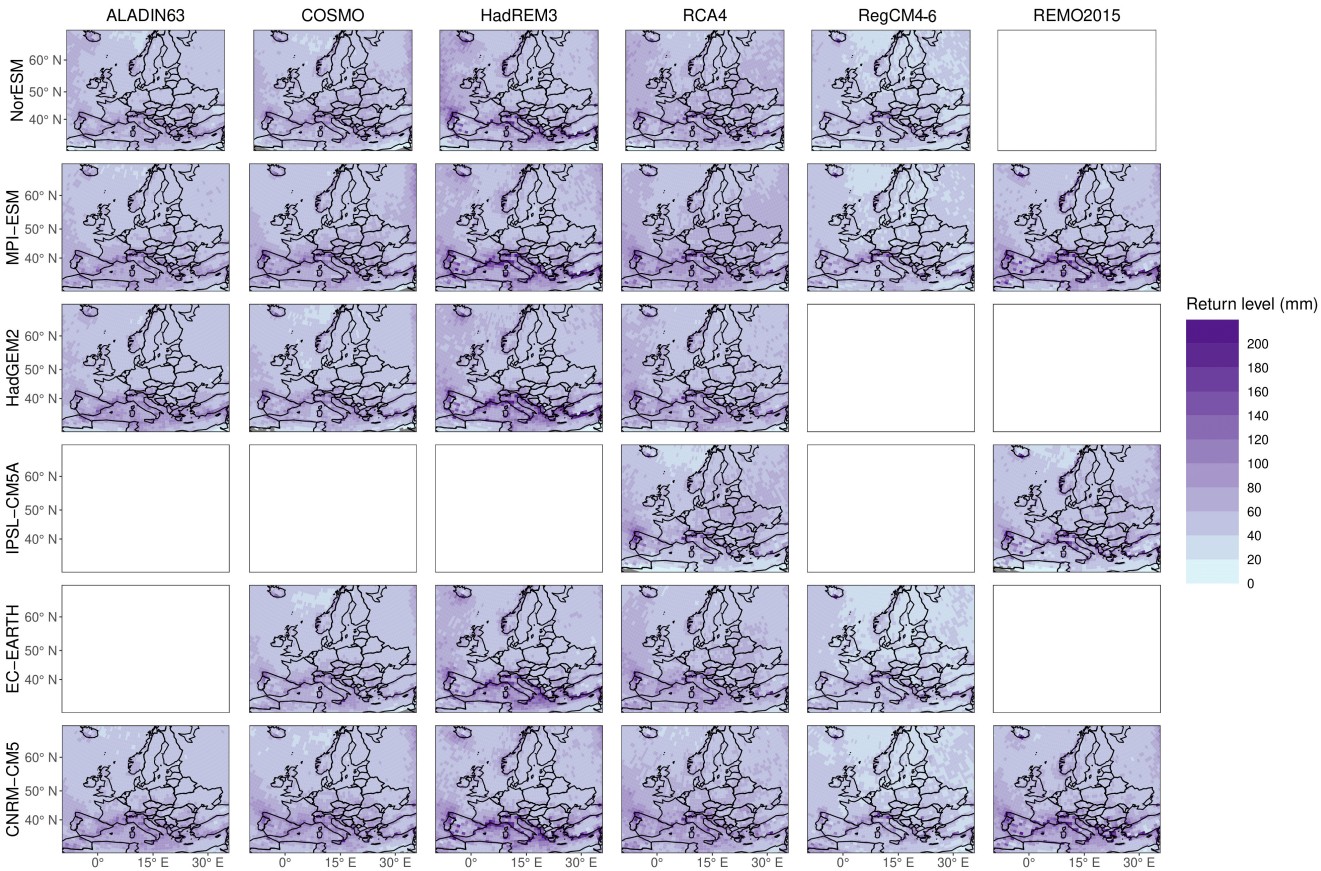

**Figure 2.** Same as Fig. 1, but for daily instead of hourly precipitation. The darkest colour shade is used for all values above 200 mm and the maximum value found is 385.8 mm.

Figure 3 summarises the spatial mean of the relative bias in the 10-year return level in six countries, for all GCM-RCM combinations for hourly and daily precipitation. Again, as noted in Sect. 2.2, biases from simulations with the same RCM are more similar than those with the same forcing GCM. For all countries except the United Kingdom (UK), almost all simulations severely underestimate the return levels of hourly precipitation by up to 60%. RCMs ALADIN63 and RegCM4-6 do so most strongly, while HadREM3-GA7-05 and REMO2015 usually feature a higher, less negative bias and sometimes even a positive bias. For the UK, about half of the simulations underestimate the return levels, while the other half overestimate them. The simulations are clearly more accurate for return levels of daily precipitation extremes than for hourly precipitation. RegCM4-6 and RCA4 gave the strongest underestimations and overestimations of the daily return levels, respectively. The biases are mostly negative for all countries except the UK.

In line with the method proposed by Vautard et al. (2021), the RCM and GCM contributions to the bias are disentangled by calculating the mean "within-GCM normalised variance" (WGNV) and the mean "within-RCM normalised variance" (WRNV) and we refer to Appendix B for technical details. The case WGNV≈WRNV indicates similar RCM and GCM contributions.



**Figure 3.** Relative bias (%) of the 10-year return levels in the EURO-SUPREME ensemble, for hourly and daily precipitation for different RCM-GCM combinations and for six countries (Belgium, The Netherlands, United Kingdom, Denmark, Germany and Finland). Both colours and numbers indicate the relative bias.




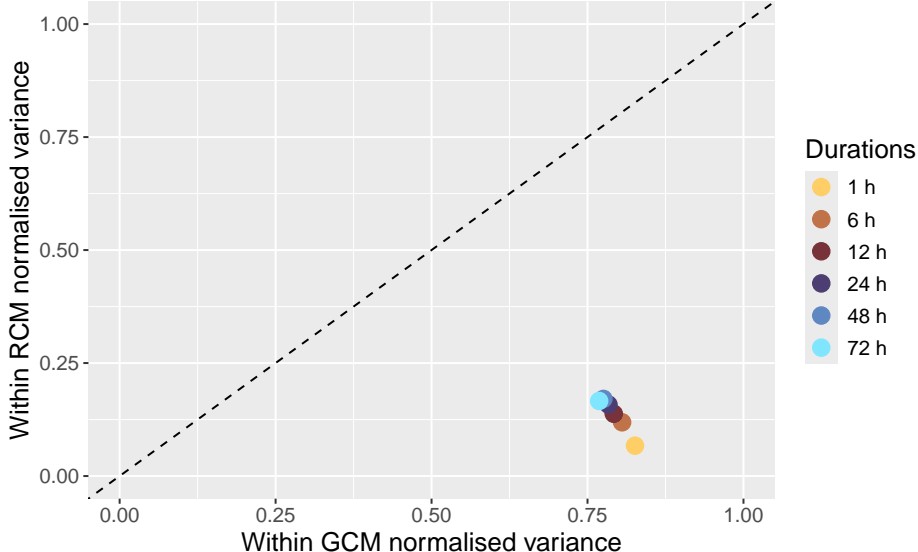

**Figure 4.** Relative variance of the historical bias of the 10-year return levels in the EURO-CORDEX ensemble, split in contribution by RCM and GCM using the mean "within-GCM normalised variance" (WGNV) and mean "within-RCM normalised variance" (WRNV), see Appendix B. Colours indicate the rainfall duration.

If WGNV>WRNV, the contribution of the RCM to the bias dominates, while WGNV<WRNV signifies that the contribution of the GCM is more significant. The ensemble averaged results for the bias (Fig. 4) clearly confirm our earlier conclusion that the RCM is by far the largest contributor to the model bias of the rainfall extremes for all durations.

## 4 Changes in EURO-CORDEX rainfall extremes under global warming

Changes in intensity and frequency of precipitation extremes between global warming levels (GWLs) of 1.5 °C and 3 °C in the EURO-CORDEX simulations were investigated. Periods corresponding to a particular GWL are identified as the first 30-year period for which the global average near-surface temperature of the forcing GCM reaches the GWL as compared to the pre-industrial period 1881–1910 (Vautard et al., 2014), and are listed in Table S4.

For the eight European "PRUDENCE" subregions (Christensen et al., 2007), we calculate (i) the change in intensity using changes in 10-year return levels, similar to Vautard et al. (2014), and, (ii) the change in frequency using changes in the annual exceedance probability of fixed reference thresholds, similar to Fischer and Knutti (2016).

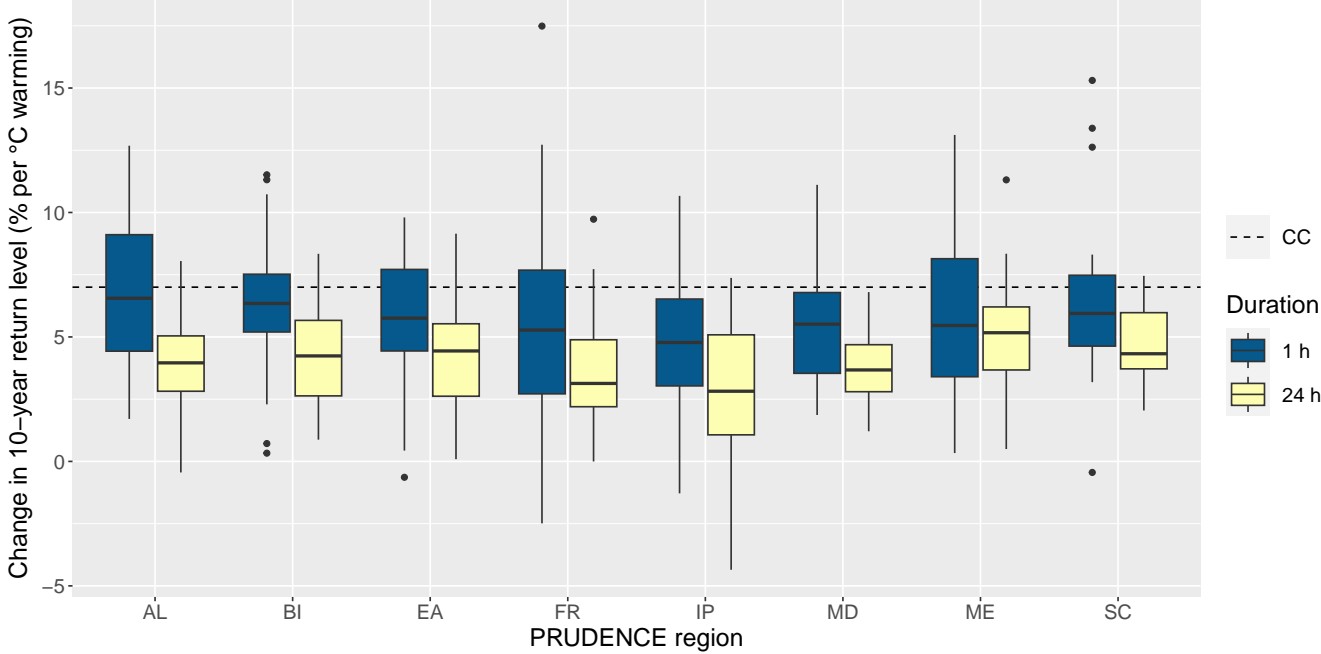

**Figure 5.** Boxplots showing the change in 10-year return level at 3°C GWL compared to 1.5°C GWL in the EURO-CORDEX ensemble, grouped by PRUDENCE region. SC stands for Scandinavia, BI for the British Isles, ME for Mid-Europe, EA for Eastern Europe, FR for France, AL for the Alps, IP for the Iberian Peninsula and MD for the Mediterranean. The colour of each boxplot indicates the rainfall duration. The CC-rate of 7% is indicated with the horizontal dashed line.

## 4.1 Change in intensity

The relative intensity difference per degree warming is obtained using the $T$-year return levels for 1.5 °C GWL (denoted $z_T^{(1.5)}$)
and 3 °C GWL (denoted $z_T^{(3)}$), as follows:

$$\text{Relative intensity change} = \frac{1}{(3\,°\text{C} - 1.5\,°\text{C})}\,\mathbb{E}\left[\frac{z_T^{(3)} - z_T^{(1.5)}}{z_T^{(1.5)}}\right], \tag{1}$$

where $\mathbb{E}[.]$ is the average over all gridpoints of the domain. Figure 5 shows the boxplots of the relative change in 10-year return levels, for hourly and daily precipitation extremes. There is a clear increase in the return-level intensities between GWL 1.5°C and 3°C. Moreover, the (median) increases for hourly precipitation are consistently larger than for daily precipitation and, in most regions, the ensemble spread is larger for hourly than for daily precipitation. The ensemble spread for the hourly
extremes is the largest over the Alps, France and Mid-Europe. The ensemble median for all regions ranges between 4.5% over the Iberian Peninsula, in line with a general drying in that region (Coppola et al., 2021), and around 7% over the Alps, agreeing with Giorgi et al. (2016). The median increase in the return level of daily precipitation extremes is lower than for hourly mean precipitation extremes, with only the upper whiskers of the boxplots crossing the 7% reference line.

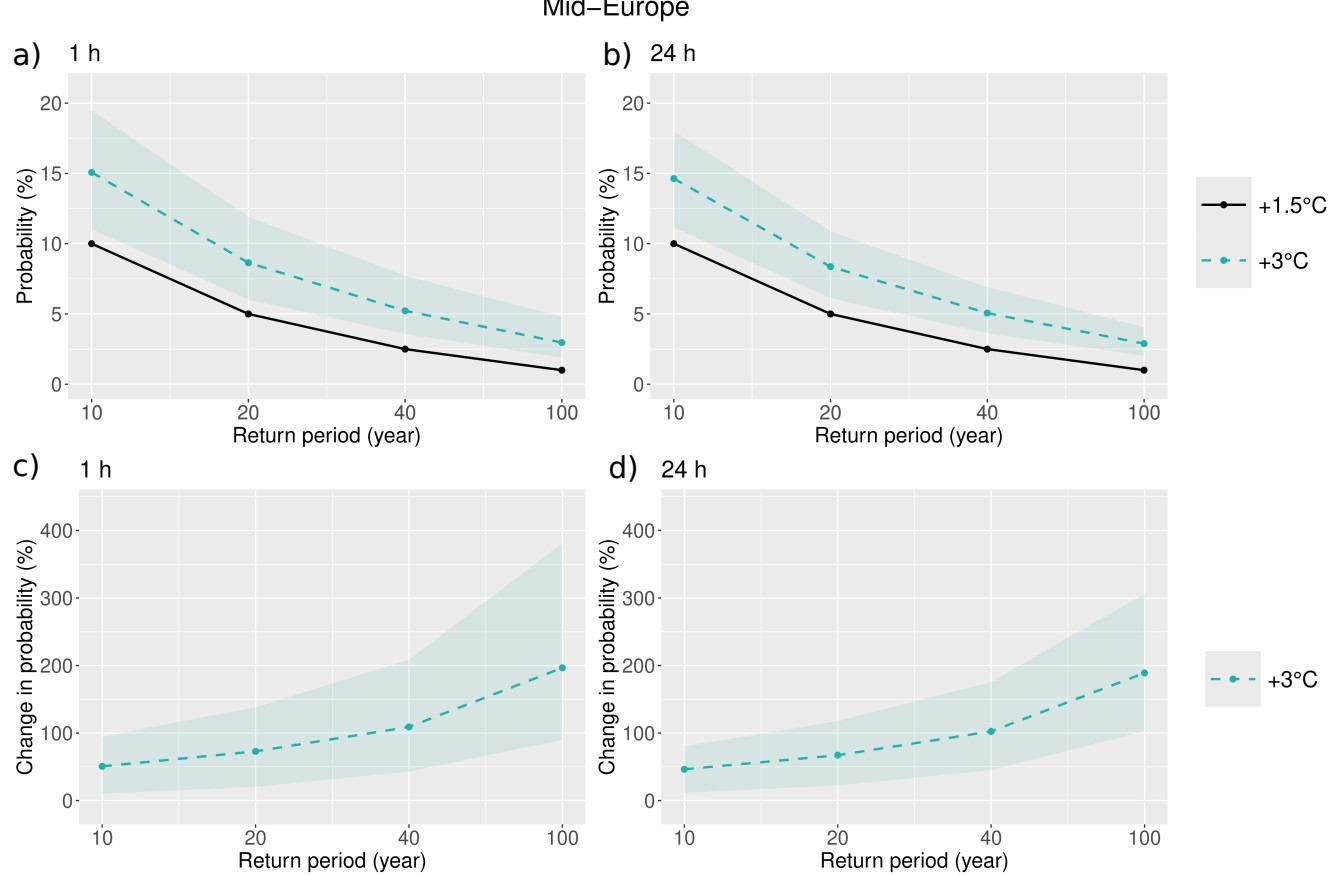

**Figure 6.** (a)-(b): annual exceedance probability (%) of the $T$-year return levels for 1.5 °C GWL ($z_T^{(1.5)}$) as a function of return period $T$ (year), for Mid-Europe. The black line shows this probability at 1.5°C GWL, and acts as a reference. Shading represents the minimum-maximum range in the EURO-CORDEX ensemble of the annual exceedance probability at 3°C GWL, and the dashed line is the associated ensemble mean. (c)-(d): the corresponding relative change of this probability between GWLs 3°C and 1.5°C. Results for hourly and daily precipitation extremes can be seen in the left and right column, respectively.

An alternative averaging approach to the domain averages of the relative intensity change (Eq. (1)) is to use the relative change of the domain-average intensities. This amounts to slightly different results, namely a higher ensemble mean in the second case, but the general conclusions still hold (Dierickx, 2024, Fig. 5.25-5.26).

## 4.2 Change in frequency

Figure 6 shows the annual exceedance probabilities (upper panels) and their changes (lower panels) for 1 h- and 24 h-precipitation extremes over Mid-Europe. In the top panels, the black dots represent the reference, which is the obvious relationship between return period $T$ (year) and annual exceedance probability of the $T$-year return level, i.e. $1/T \times 100$ (%).





The green shading and dotted line, on the other hand, correspond to the domain-averaged exceedance probability at 3 °C GWL, calculated by how often the 3 °C GWL annual maxima are expected to exceed the 1.5 °C return levels. More specifically:

$$\text{Annual exceedance probability of } z_T^{(1.5)} \text{ at 3 °C} = \left[1 - G^{(3)}\left(z_T^{(1.5)}\right)\right], \qquad (2)$$

with $G^{(3)}$ the fitted GEV-distribution at 3 °C GWL.

The relative changes of the annual exceedance probability between 3 °C and 1.5 °C are shown in the lower panels of Fig. 6. For example, for a return period of $T = 100$ years, the return level of hourly precipitation at 1.5 °C GWL is three times more likely to be exceeded at 3 °C GWL according to the ensemble mean. The models unanimously agree on a frequency increase for all return periods. The longer the return period, the higher the relative frequency increase for both hourly and daily

precipitation. The ensemble spread, represented by the shading, also increases with the return period.

## 5   Example applications

Here we show how EURO-SUPREME can be used as a benchmark dataset for the evaluation of extreme precipitation simulated by high-resolution convection-permitting models (CPMs). We provide an example by using CPM runs over Belgium, performed in the framework of the CORDEX.be project (Termonia et al., 2018; Van de Vyver et al., 2021), and comparing

them to EURO-SUPREME and ground station data. We consider three historical CPM simulations: ALARO-0 (4 km, 1 run) and COSMO-CCLM (2.8 km, two runs with COSMO v5.0 and v6.0) of which more technical details can be found in Termonia et al. (2018, Table 2). The ground-station data consist of hourly precipitation observations at 176 stations in Belgium (Fig. S1). The CPM biases are indicated by blue squares in Fig. 7a and can be compared with the EURO-SUPREME biases, shown as boxplots. The CPMs significantly reduce the bias wrt EURO-SUPREME for the hourly extremes as they fall (well) outside

the ensemble quartile of EURO-SUPREME. This reduction gradually disappears as rainfall duration increases, and for 24-h extremes, the CPMs no longer add value.

Producing an acceptable bias may be considered a necessary but not a sufficient condition for adequate confidence in future projections. Enhanced reliability may come from the consideration of physically-meaningful performance metrics, such as the correct simulation of the diurnal cycle, the temperature and humidity dependency or the spatial structure (Westra et al.,

2014; Cortés-Hernández et al., 2016; Van de Vyver et al., 2021; Dierickx, 2024). Here, we investigate whether the models can adequately reproduce the relationship between orography and extreme precipitation. Figure 7b shows the Spearman's rank correlation between the 10-year return levels and elevation. The low correlations associated with the observations (black dots) with duration $d \leq 12$ h indicate a very limited orographic influence. The models, on the contrary, feature a much higher correlation, which is remarkably higher for CPMs. Consistent with this, it was noted that orographic effects in the Alps are not

fully accounted for in CPMs (Dallan et al., 2023), although the elevation range over the (southern) Belgium is much smaller than over the Alps.





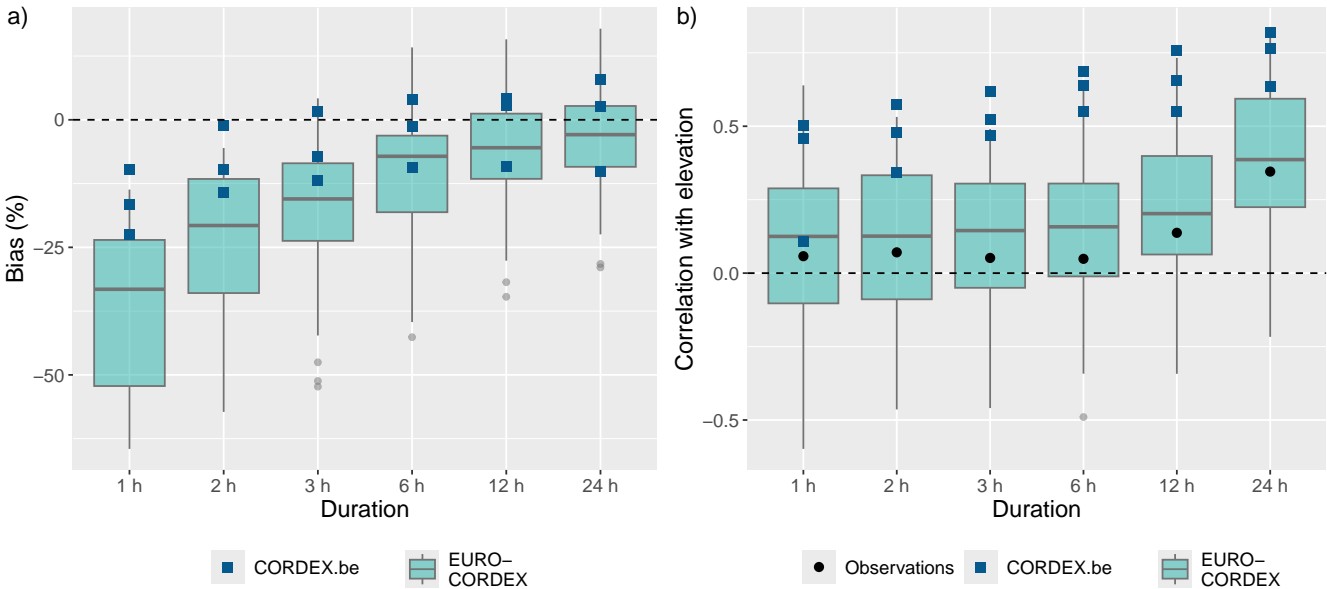

**Figure 7.** a) Relative bias (%) in the 10-year return levels based on EURO-SUPREME (boxplots) and the three high-resolution CPM runs of CORDEX.be. b) Spearman's rank correlation of elevation versus 10-year return levels for EURO-SUPREME (boxplots), observations (black dots) and the CORDEX.be CPM runs (blue squares).

## 6 Data availability

The EURO-SUPREME dataset is made available under CC-BY-4.0 licence through the German Climate Computing Centre (DKRZ) and can be accessed via https://doi.org/10.26050/WDCC/EURCORDEX_prec (Van de Vyver et al., 2024).

The high-resolution convection-permitting simulations over Belgium used in Sect. 5 were developed in the CORDEX.be project (Termonia et al., 2018) and are available at https://www.geo.be/catalog/details/22dc62f0-ca7f-11ee-9a71-847b573ec00f? l=en.

The hourly station observations of precipitation used in Sect. 5 come from several Belgian monitoring networks managed by:

– Royal Meteorological Institute (RMI). Data available at https://opendata.meteo.be/download.

– Vlaamse Milieumaatschappij (VMM). Data available at https://waterinfo.vlaanderen.be/

– Service Public de Wallonie (SPW) - Mobilité et Infrastructures. The data may be requested from the Service public de Wallonie - Mobilité et Infrastructures, Direction de la Gestion hydrologique, PEREX, Rue Del'Grête, 22, 5020 NAMUR (Daussoulx).





– Bruxelles Environnement - Leefmilieu Brussel. The IBGE rainfall monitoring network is set up by the Ministry of the Brussels-Capital Region and managed since 2007 by the IBGE (Dehem et al., 2010; Journée et al., 2014) and the data can be obtained upon request at https://environnement.brussels.

## 7   Code availability

The processing code can be obtained on request from the corresponding author.

## 8   Discussion and Conclusion

The EURO-SUPREME dataset introduced here contains sub-daily extreme precipitation events derived from a 35-member EURO-CORDEX 0.11° ensemble that includes both historical and future (RCP8.5) scenarios. The dataset contains accumulated precipitation depths over periods ranging from 1 hour to 72 hours, for each grid point of the EURO-CORDEX domain. Among the many potential uses of this dataset, we illustrated how the dataset can serve as a benchmark for evaluating high-resolution CPMs.

Validation of the models within the EURO-SUPREME dataset using observations shows that, for all countries investigated, EURO-CORDEX models produce return levels of daily rainfall extremes more accurately than for hourly extremes and biases are mainly determined by the RCM. Hourly extremes are often severely underestimated, while daily extremes are sometimes overestimated. The validation results are reasonably similar between countries, with the UK differing slightly more from the others. The significant model biases motivate the need to apply appropriate bias-adjustment techniques (Schmith et al., 2021; Van de Vyver et al., 2023).

The EURO-CORDEX models unanimously project an increase in the 10-year return levels between GWLs 1.5 °C and 3 °C. On a relative basis, return levels for shorter rainfall durations will increase more than those for longer durations. There is, however, a large model spread in the relative increase of the return levels, which is fairly similar between the PRUDENCE regions. For hourly precipitation, the ensemble median for each region lies slightly below the CC rate, while it is systematically smaller for daily than for hourly extremes. Moreover, the frequency of exceeding the return levels at 1.5 °C GWL will rise at 3 °C GWL. A larger relative increase in frequency is seen for more intense extreme precipitation events, cfr. Fischer and Knutti (2016).

As in Berg et al. (2019) and Poschlod et al. (2021), simulated precipitation extremes were validated by comparing 10-year return values between simulations and observations. Due to the limited ensemble sizes per model, the uncertainties may be large (Wehner, 2010). Poschlod and Ludwig (2021), for instance, show that the impact of internal variability is reduced within a 50-member ensemble of single RCM initial-condition simulations. On the other hand, the EURO-SUPREME dataset may provide a more complete picture of model errors using a 35-member multi-model ensemble consisting of 27 different GCM-RCM pairs, which is also considerably larger than the 9-member ensemble of Berg et al. (2019). In addition, uncertainty of observed return levels may be determined by the statistical method used to obtain national precipitation statistics which may





differ from country to country. However, a comparative study in Poschlod et al. (2021) shows that the observed return levels show smooth transitions between the borders of neighbouring countries, and the reference can therefore be considered reliable. The recently released GSDR-I dataset (Pritchard et al., 2023) of extreme indices of observed sub-daily precipitation may be used to extend our model-validation efforts.

The EURO-SUPREME dataset contains only annual maxima and no return levels. This offers more flexibility for the user, for example when choosing the fitting period or performing more advanced assessments, such as intensity-duration-frequency (IDF) analysis (Hosseinzadehtalaei et al., 2020) or extremal dependence over space based on variogram analysis (Cooley et al., 2006; Cortés-Hernández et al., 2016; Van de Vyver et al., 2021; Yang et al., 2023). In fact, there are numerous user-friendly software packages for extreme-value analyses (Belzile et al., 2023).

For the selection of extreme events, we chose annual maxima of accumulated precipitation, because there is a well-founded theory for modelling them statistically (Coles, 2001; Beirlant et al., 2004). The alternative to the block-maxima method, the peaks-over-threshold method, includes more data and can therefore reduce uncertainty but it has the disadvantage that the choice of the optimal threshold (e.g. in the form of a high quantile of non-zero precipitation) is subjective and can vary spatially. On the other hand, there is a strong theoretical relationship between block maxima and threshold models (Coles, 250 2001). Also, the uncertainty of the block maxima method can be reduced by combining data spatially in a spatial regression model for extremes (Casson and Coles, 1999; Cooley et al., 2007; Van de Vyver, 2012).

In addition to benchmarking, the simulated return levels of daily precipitation can be used in areas with little data (although with some caution for orographically complex regions) because validation shows a fairly good agreement with observations, an idea already suggested by Poschlod et al. (2021). Assuming the models are equally plausible, the multi-model ensemble 255 average can be considered, which is generally known to provide better estimates than a single model (Tebaldi and Knutti, 2007). Alternatively, a weighted multi-model simulation based on model quality or climate sensitivity can be taken into account (Knutti et al., 2017; Massoud et al., 2023). Finally, extreme-event attribution studies are usually made more robust by complementing observations with multi-model simulations (Philip et al., 2020; Tradowsky et al., 2023; Kimutai et al., 2024) such as our EURO-SUPREME dataset.

**Appendix A: Statistical modelling of annual maxima with the generalised extreme value distribution**

The $T$-return level, $z_T$, corresponds to the value that is expected to be exceeded on average once per $T$ years, and can be computed by means of an extreme-value distribution. Annual maxima are commonly modelled with the generalised extreme value (GEV) distribution where the cumulative probability distribution has the form (Coles, 2001):

$$G(z) = \exp\left\{-\left[1 + \xi\left(\frac{z-\mu}{\sigma}\right)\right]_+^{-1/\xi}\right\}, \qquad \text{with } (v)_+ = \max(0, v), \tag{A1}$$

where $\mu$ is the location parameter, $\sigma > 0$ is the scale parameter and $\xi$ is the shape parameter. For each grid point and rainfall duration, we fit the GEV distribution to the annual maximum series using the probability-weighted method (Hosking et al., 1985). This estimator has the advantage of low variance and is less sensitive to outliers compared to the maximum-likelihood





estimator, especially for small sample sizes (Martins and Stedinger, 2000). The output of the GCM-RCM simulations using multiple GCM members (see Table 1) was combined into a long time series for each grid point.

After the GEV parameters are fitted to an annual maximum series, the return levels can be computed by inverting the equation: $G(z_T) = 1 - 1/T$, with $G$ given in Eq. (A1). We get:

$$z_T = \mu - \frac{\sigma}{\xi} \left\{ 1 - \left[ -\log\left(1 - \frac{1}{T}\right) \right]^{-\xi} \right\}. \tag{A2}$$

**Appendix B:  Variance decomposition by GCM and RCM**

The contributions of the GCM and RCM to the total simulation error can be disentangled by calculating the mean "within-GCM

normalised variance" (WGNV) and the mean "within-RCM normalised variance" (WGRV), similar to Vautard et al. (2021). For each GCM $i$ and RCM $j$, the following definitions are used for decomposing the variance of the historical bias $B$:

$$\mathrm{WGNV}_i = \frac{1}{N_{\mathrm{country}}} \sum_{\mathrm{country}\,k} \frac{1}{V_{tot,k}} \sum_{\mathrm{RCM}\,j} \left(B_{ij,k} - \overline{B}_{i,k}\right)^2 \tag{B1}$$

$$\mathrm{WRNV}_j = \frac{1}{N_{\mathrm{country}}} \sum_{\mathrm{country}\,k} \frac{1}{V_{tot,k}} \sum_{\mathrm{GCM}\,i} \left(B_{ij,k} - \overline{B}_{j,k}\right)^2, \tag{B2}$$

where $B_{ij,k}$ is the bias of RCM $j$ downscaling GCM $i$ in country $k$. Furthermore, the mean bias per RCM and GCM and the

normalisation factor per country are given by:

$$\overline{B}_{i,k} = \frac{1}{N_{\mathrm{GCM}_i}} \sum_{\mathrm{RCM}\,j} B_{ij,k} \tag{B3}$$

$$\overline{B}_{j,k} = \frac{1}{N_{\mathrm{RCM}_j}} \sum_{\mathrm{GCM}\,i} B_{ij,k} \tag{B4}$$

$$V_{tot,k} = \frac{1}{N} \sum_{\mathrm{RCM}\,j} \sum_{\mathrm{GCM}\,i} \left(B_{ij,k} - \overline{B}_k\right)^2, \tag{B5}$$

where $N_{\mathrm{GCM}_i}$ is the number of simulations forced with GCM $i$, $N_{\mathrm{RCM}_j}$ is the number of simulations with RCM $j$ and $N$ is the

total number of (available) simulations. Note that this definition is slightly different from the one used in Vautard et al. (2021), who use a single normalization factor for all countries (or subdomains) and may therefore exceed the value of one.

If, for example, the bias is solely determined by the GCM $i$, the associated $\mathrm{WGNV}_i$ would be zero, as $B_{ij,k}$ would not depend on the RCM $j$ ($B_{ij,k} = \overline{B}_{i,k}$ for every RCM $j$). Similarly, if the bias only depends on the RCM $j$, the $\mathrm{WRNV}_j$ would be zero. The maximum value for $\mathrm{WRNV}_j$ and $\mathrm{WGNV}_i$, on the other hand, can be easily proven to be one. These quantities

can thus be used to see whether the GCM or the RCM dominates the bias.

Finally, we calculate the mean WGNV and mean WRNV as the weighted averages over all GCMs and RCMs, respectively:

$$\mathrm{WGNV} = \sum_{\mathrm{GCM}\,i} \left(\frac{N_{\mathrm{GCM}_i}}{N}\right) \mathrm{WGNV}_i \tag{B6}$$

$$\mathrm{WRNV} = \sum_{\mathrm{RCM}\,j} \left(\frac{N_{\mathrm{RCM}_j}}{N}\right) \mathrm{WRNV}_j. \tag{B7}$$



*Author contributions.*  AD, BVS and HVDV wrote the manuscript with contributions from all co-authors. Formal data analysis, validation,
interpretation, and visualization were performed by AD, under supervision of WD, BVS, SC and HVDV. BVS, SC and HVDV conceptualized
the study. HVDV, BVS and LD did the data processing. HVDV liaised with supporting organisations (notably DKRZ). PT, SC and BVS were
responsible for the project administration.

*Competing interests.*  The authors declare that they have no conflict of interest concerning the publication of this article.

*Disclaimer.*  Publisher's note: Copernicus Publications remains neutral with regard to jurisdictional claims made in the text, published maps,
institutional affiliations, or any other geographical representation in this paper. While Copernicus Publications makes every effort to include
appropriate place names, the final responsibility lies with the authors.

*Acknowledgements.*  This research was supported by the Belgian Science Policy (BELSPO) under Contract B2/223/P1/CORDEXbeII.
We also acknowledge the Earth System Grid Federation infrastructure, an international effort led by the U.S. Department of Energy's
Program for Climate Model Diagnosis and Intercomparison, the European Network for Earth System Modeling and other partners in the
Global Organisation for Earth System Science Portals (GO-ESSP).





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
