# Peer review of "EURO-SUPREME: sub-daily precipitation extremes in the EURO-CORDEX ensemble"

_Earth System Science Data, 2025_

## Author Comment (AC1)

**Response to Anonymous Referee #1**

August 27, 2025

**General comment**

The dataset description is excellent, and the dataset will prove to be very useful. The collection is in principle redundant, as all data are already available on ESGF. I do, however, find it very useful to collect this processed dataset in one place. Annual maxima of varying duration should be a very useful intermediate for plenty of potential studies.

**Reply.** We thank the reviewer for the overall positive assessment of the dataset, and the useful comments and suggestions.

**Specific comments**

- Would it be possible in the future to supplement the current data collection with relevant files from other EURO-CORDEX5 simulations with sub-daily precipitation data? Some simulations have published 3-hourly data, which may complete any analysis of temporal resolution below this period. In case you have plans to do this, please mention it in the text. -Otherwise you may ignore this comment.

  **Reply.** We indeed do not plan to add 3-hourly data for scientific reasons. Firstly, because we are primarily interested in precipitation durations of 1-3 hour for benchmarking experiments with convection-permitting simulations. Secondly, with regard to longer rainfall durations, sliding 24-hour accumulations will lead to an underestimation of 24-hour extremes.

- Please employ the alternative method mentioned in l155-157 instead of the one currently used. It is good practice to average additive quantities before taking ratios, in order to avoid undue weight for points with very small numbers in the denominator. Please revise the text accordingly

  **Reply.** We have used the suggested alternative method and arrived at similar conclusions, which will be included in the next submission. Equation 1 will be changed from

  $$\text{Relative intensity change} = \frac{1}{(3\,^{\circ}\text{C} - 1.5\,^{\circ}\text{C})}\,\mathbb{E}\left[\frac{z_T^{(3)} - z_T^{(1.5)}}{z_T^{(1.5)}}\right], \tag{1}$$

  to

  $$\text{Relative intensity change} = \frac{1}{(3\,^{\circ}\text{C} - 1.5\,^{\circ}\text{C})}\,\frac{\mathbb{E}\left[z_T^{(3)} - z_T^{(1.5)}\right]}{\mathbb{E}\left[z_T^{(1.5)}\right]}. \tag{2}$$

  Furthermore, Figs. 4 and 7 will be modified to use the values of relative intensity change calculated using the new method.

- Please mention exactly how you find the 1.5 GWL and 3 GWL periods (probably trivial, but nice for the documentation of methods). Your method for calculating change per degree GWL is a bit unconventional; please discuss how big an effect this has relative to the more frequently used one of taking changes from historical to end-of-century and dividing by whatever global warming happens between those two periods. I

do see advantages in you method related to avoidance of extreme periods in very sensitive simulations, but please add some discussion.

**Reply.** Periods corresponding to a particular GWL are identified as the first 30-year period for which the global average near-surface temperature of the forcing GCM reaches the GWL as compared to the pre-industrial period 1881–1910. The methodology with fixed global warming levels is supposed to be more robust than using fixed time periods as some uncertainty from the climate sensitivity of the GCMs is removed. The explanation of how and why GWL periods are determined can be found in Vautard *et al.* (2014) to which we will refer in the next submission. In the text, we add: *"The periods are taken from Vautard* et al. *(2014) and are listed in Table S4. Vautard* et al. *(2014) argue that this methodology with fixed global warming levels is more robust than using fixed time periods as some uncertainty from the climate sensitivity of the GCMs is removed."*

- Please consider using a more diverse colour palette in figs 1 and 2. It is currently very hard to distinguish levels.

**Reply.** We will use a more diverse colour palette, `batlow`, which is a perceptually uniform, perceptually ordered and colour-vision-deficiency friendly (see https://www.fabiocrameri.ch/batlow/). As a result, for example, the relatively high values simulated by HadREM3 for the Mediterranean region are much more clearly visible.

---

## Author Comment (AC2)

**Response to Anonymous Referee #2**

August 27, 2025

**Summary**

First of all thanks to the authors for processing and compiling this useful dataset on annual maximum precipitation amounts for a subset of the EURO-CORDEX RCM ensemble and the comprehensive and extensive accompagnying study, which demonstrates the potential usefulness of the dataset. Despite the fact that the data is basically available through the Earth System Grid Federation data nodes, the data processing and compilation and sharing as FAIR open access research data makes total sense. The manuscript is well written, the dataset is well prepared and fits the scope of the journal. Some open issues as to the construction of the ensemble, the processing of the dataset and the presentation through the data descriptor paper albeit remain.

**Reply.** We thank the reviewer for the overall positive assessment and the relevant comments and suggestions.

**General comments**

The dataset, or rather the data product, is novel and useful to and usable by the community. Based on information provided the data product could be reproduced, if needed; see my comments below, a little bit more detail would be desirable. The data product is presented with enough context to existing literature; with some sections though, more references to existing CORDEX analysis may be useful. The manuscript supports the dataset well with very useful examples.

The dataset quality is fine, the dataset DOI works well, data meet FAIR principles. The data is findable and accessible (after free user registration) through the long-term WDCC storage and dissemination infrastructure, uses compressed netCDF-4 as an interoperable data format, complete with meta data and provenance information as well as version control. Common standards are met. Some notes and recommendations on dataset processing and refinements are given below. The dataset itself is of high quality. There does not seem to be any inconsistency between the paper manuscript and the dataset.

The dataset is useful and usable; again some proposition is made below to increase this further by regridding the data to a common grid. The manuscript is properly structured and clearly written. Methods are described in more detail in a useful appendix. Visual material are OK, some minor comments are given below.

**Reply.** We agree that the dataset could be further improved based on these suggestions, and have therefore generated an extended dataset and requested a new DOI from DKRZ so that it can be published online. In particular, we have now supplemented the original dataset with regridded data into a common grid. We have also added evaluation runs (whenever available) and useful variables such as "surface height" (orog) and "land_area_fraction" (sftlf) to all NetCDF files.

**The current upload process has started at DKRZ, and we expect the new DOI for the extended dataset to be ready in September 2025. Meanwhile, the extended dataset is already available to reviewers via Google Drive.**

**Specific comments**

- P/P2/L44ff: Given the validation and usage examples you provide later, perhaps it makes sense to emphasize that the data is especially useful for extreme value statistics.
  **Reply.** We now explicitly refer to extreme value statistics.

- P/P3/L70ff: Perhaps mention that there are many more ensemble members available, but you were after sub-daily data, which are provided by much less ensemble members of the CORDEX-CMIP5 simulations. Maybe also address here, that the CORDEX-CMIP5, other than CORDEX-CMIP6, was not based on some balanced matrix design as described in Katragkou et al. (2024, BAMS), who you cite. So I wonder what motivates eventually the ensemble subset the dataset covers? The EURO-SUPREME dataset does not contain evaluation run results, but especially for using the data for benchmarking, e.g., CPRCM simulations or other validation purposes. Especially for the Section 3 evaluation, this evaluation run dataset might also be helpful. Would it make sense and could this still be added? For completeness also the land-sea masks ("sftlf" variable) and the orography ("orog" variable) data should be provided.
  **Reply.** We will provide a more complete description of the selection of the EURO-CORDEX ensemble members.

  Firstly, we mention that of the total 69 GCM-RCM combinations in the 0.11° ensemble, only 27 provide precipitation data at an hourly temporal resolution.

  Secondly, we explain that a drawback of using a relatively small number of GCM-RCM pairs is that their selection was based solely on data availability, rather than on the principles of the EURO-CORDEX Balanced Ensemble Design (Katragkou *et al.*, 2024), which takes into account additional factors, such as physical plausibility, representativeness of future climate distributions, and the statistical independence of the models.

  Thirdly, as mentioned above in *"General comments"*, we have added evaluation runs, but they are only available for four RCMs: ALADIN63, COSMO-crCLIM, HadREM3-GA7-05 and RCA4.

  Finally, a land-sea mask ("sftlf" variable) and the orography ("orog" variable) have been added to all NetCDFs.

- P/P3/L75: You mention the data is on different grids, which the reviewer can confirm from personal experience. However, given you want to provide a dataset most easy to use, would it not be better to provide the data ona common grid, e.g., a curvilinear EUR-11 grid, based on the original grid specification of EUR-11 (now EUR-12)? You may regrid using nearest neighbour resampling. This seems to me a weakness of the dataset provided.
  **Reply.** As already mentioned in the *"General comments"*, all simulations were now regridded to a common regular $0.11° \times 0.11°$ grid with spatial coverage $28°N-70°N$ and $13°W-35°E$. This was done using nearest neighbour remapping, using the CDO-command `remapnn`. The regridded data was added to the data on the original grid.

- P/P3/L76ff: Perhaps structure the Section 2.2 a bit more, separating the variables desciption, data formats, data access, etc. from each other. In L92ff your systematic pre-processing / quality checking steps may be described in more detail; right now it reads as if on accident outliers, grid point storms, etc. were detected. Please indicate whether you applied a systematic check.
  **Reply.** Thank you for this useful suggestion. We have restructured and extended the description and divide it into three subsections: Sec. 2.2.1 *"Description of the files and variables"*, Sec. 2.2.2 *"Outliers"*, and Sec. 2.2.3 *"Preliminary visualisation of the dataset"*.
  We systematically checked the hourly precipitation data for unrealistic values by identifying exceedances above a threshold value of 100 mm/h, that could be attributed to the occurrence of non-physical "grid point storms".

- P/P3/L77: Maybe introduce the block maxima method, later on discussed in the Sect.8.
  **Reply.** This is being changed.

- P/P4/L99: In addition to Fig.1 and Fig.2 perhaps provide an example dataset visualisation, like the 24h maximum annual precipitation for any given year, so the reader gets an idea of the nature of the dataset. Starting off with the return levels, i.e., a drived quantity based on the originally provided data could come after that.
**Reply.** A specific year of an historical simulation (COSMO-crCLIM) is plotted for hourly maxima, and we plan to show this in Fig. 1, just before the existing return level maps.

- P/P5/L198ff/Sect.3: Would it also make make sense to cite some of the many studies which have validated and evaluated the EURO-CORDEX CMIP5 RCM ensemble precipitation dataset? In light of the previous Fig.1/2 it makes sense to evaluate the return levels. However, given the seemingly sketchy reference data and the spatial aggregation, would it not make more sense to compare to the very same diagnostic as you provide, annual maximum precipitation per grid element per duration class, and assess typical bias measures? For example for PRUDENCE regions, which you use in Section 5 anyhow? Because given your return level evaluation yields results with larger biases, the reader might question the dataset usefulness. I do think however the dataset is very useful, and it is know that RCMs exhibit biases with precipitation. At the same time, the return level comparison allows you to compare the statistical properties of the historical run vs observations. So without using an evaluation (ERA-Interim driven) simulation based precipitation diagnostic a direct comparison is difficult as well.
**Reply.** We add references to standard works on the validation of EURO-CORDEX simulations (e.g. Kotlarski et al., 2014; Rajczak and Schär, 2017; Vautard et al., 2021; Barnes et al., 2024).

The advantage of the diagnostics proposed above, which consists of comparing the annual maxima of simulations and observations rather than comparing the corresponding $T$-year return levels, is that it involves less uncertainty, especially for (very) long return periods $T$. However, $T = 10$ years is quite common and acceptable for an evaluation, as previously illustrated in Berg *et al.* (2019) and Poschold *et al.* (2021). Furthermore, the reviewer's suggestion may be a valuable option only if the observed $d$-hourly precipitation extremes are directly available but this is, however, not the case. We must therefore limit ourselves to national statistics.

We also compared the 10-year return levels from the evaluation runs (downscaled from ECMWF-ERAINT) with those from the observations and obtained quantitatively similar results per model in reproducing subdaily rainfall extremes. This will be briefly reported in the next submission.

- P/P7/L129f: You should make clear you still refer to the return level biases.
**Reply.** This is being changed.

- P/P9/L135ff/Sect.4: Would this section perhaps fit better under the Section 5 with application examples as the goal of the paper is primarily on the dataset you present and not the derived climate change analysis?
**Reply.** Thank you for the suggestion. Sec. 4 *"Example applications"* of the revised manuscript are split into Sec. 4.1 *"Benchmarking CPRCMs"* and Sec. 4.2 *"Changes in EURO-CORDEX rainfall extremes under global warming"*, with the latter section corresponding to Sec. 4 of the original manuscript.

- P/P12/L171ff: This is an interesting application example. Only I would not consider here the CORDEX-SUPREME the benchmark dataset, it is used in a benchmarking experiment, but the benchmark dataset is the observational reference data from the synop stations.
**Reply.** Indeed, this may cause confusion. We have therefore replaced "benchmark" with "benchmarking experiment".

- It may not be required by the journal, but as you put a lot of emphasis on the analysis through the return levels, it would be interesting to the reader to also have the code available through a long-term public git repository.
**Reply.** We extend Sec. 6 *"Code availability"* by specifying which R-packages and functions can be used to fit the GEV-distribution and then calculate the corresponding return levels. Finally, the codes were now made public via the GitHub-repository https://github.com/anodieri/extreme-precipitation-figures.

**Technical corrections**

- P: Out of curiosity, what does "SUPREME" stand for? Does it have a meaning?
  **Reply.** We were not very clear about that. It stands for *"EURO-SUbdaily PRecipitation extrEMEs"* and is being stressed now in the Abstract.

- P/P1/L6: replace: "0.11" (redundant with following and information is missing) with "regional climate model (RCM)"
  **Reply.** This is being changed in the Abstract.

- P/P1/L6: rephrase: "(coupled to CMIP5)" to "(downscaling CMIP5 GCMs)"
  **Reply.** This is being changed in the Abstract.

- P/P1/L6: rephrase: "precipitation depths" to "precipitation amounts"
  **Reply.** This is being changed in the Abstract.

- P/P1/L7: rephrase: "More specifically ... by a" to "Specifically, data are based on a 35-member RCM ensemble ..."
  **Reply.** This is being changed in the Abstract.

- P/P1/L8: add: "EURO-CORDEX EUR-11 (0.11°) domain"
  **Reply.** This is being changed in the Abstract.

- P/P1/L20: missing: reference to last sentence.
  **Reply.** We make a reference to Rajczak and Schär (2017).

- P/P2/L22: remove: "urban water management", just "water management"
  **Reply.** This is being changed in the Introduction.

- P/P2/L32: rephrase: call the models CPRCMs, as you refer to RCMs at km-scale resolution
  **Reply.** "CPMs" is changed to "CPRCMs".

- P/P2/L33f: rephrase: the computational resources needed are bigger, this is presumably what you mean; also cite Schaer et al. (2020, https://doi.org/10.1175/BAMS-D-18-0167.1), with an overview of computational demands of CPRCM simulations
  **Reply.** We will rephrase this as follows: *"However, the computational resources are greater than that of coarse-resolution RCMs"*. We also cite Schär *et al.* (2020).

- P/P2/L34f: add: also the length of the simulations (so far) is rather short, aside from ensemble and domain size
  **Reply.** We add: *"Also, the length of the simulations (so far) is quite short apart from the ensemble and domain size."*

- P/P2/L35ff: there are also other ways to demonstrate the CPRCM added value, I think this is a misleading motivation; the fact is that the CPRCM may simulate extreme precipitation more accurately, albeit they may be of limited use due to the constraints you mention
  **Reply.** We understand that this sentence may be misleading and has little added value. We will therefore remove it.

- P/P2/L39f: mention the CPRCM benchmark objective later on, in line 44f
  **Reply.** This will be replaced as suggested.

- P/P2/L44ff: align the listing of purposes with the overview in the abstract, to make the motivation for the dataset very clear
  **Reply.** We aligned the list of purposes in the abstract with the order that is used in the text.

- P/P3/L63f: explain the CORDEX acronym, also it is not a "framework" but a WCRP "project", cite Gutowski et al. (2016, https://doi.org/10.5194/gmd-9-4087-2016)
  **Reply.** We add: *"... Coordinated Regional Climate Downscaling Experiment (CORDEX) project of the World Climate Research Program (WCRP) (Gutowski Jr. et al., 2016)."*

- P/P3/L84f: It seems you follow the data reference syntax and filename nomenclature as used by CORDEX, which I think is very good. Perhaps you want to indicate this.
  **Reply.** We will mention the use of the CORDEX nomenclature.

- P/P6,7/Fig1,2: Please use a more differentiating color-scale. Despite adjusting the colorbar range between 1h and 24h annual precipitation maxima return levels, the colormap should be the same for the same kind of variable.
  **Reply.** We will redraw Figs. 2–3 using colour palette `batlow`, which is a perceptually uniform, perceptually ordered and colour-vision-deficiency friendly (see https://www.fabiocrameri.ch/batlow/). As a result, for example, the relatively high values simulated by HadREM3 for the Mediterranean region are much more clearly visible.

- P/P6,7/Fig1,2: Please indicate that you do not show the entire EUR-11 CORDEX domain for readers not so familiar with the CORDEX RCM ensemble.
  **Reply.** We indicate in the caption that we show a large part of the EUR-11 CORDEX domain.

- P/P10/Fig.5 caption: return level of what?
  **Reply.** We add to the caption of Fig. 7: *"10-year return levels of d-hourly precipitation"*

- P: partly there is a mix of British and American English spelling. Please double-check this throughout.
  **Reply.** We will double-check the manuscript and opt for British English.

- P/P12/L190: remove: "over the (southern)" − > "(southern)"
  **Reply.** This is being removed.

---

## Author Response (AR1)

**Author's response**

September 15, 2025

**1 Response to Anonymous Referee #1**

**General comment**

The dataset description is excellent, and the dataset will prove to be very useful. The collection is in principle redundant, as all data are already available on ESGF. I do, however, find it very useful to collect this processed dataset in one place. Annual maxima of varying duration should be a very useful intermediate for plenty of potential studies.

**Reply.** We thank the reviewer for the overall positive assessment of the dataset, and the useful comments and suggestions.

**Specific comments**

- Would it be possible in the future to supplement the current data collection with relevant files from other EURO-CORDEX5 simulations with sub-daily precipitation data? Some simulations have published 3-hourly data, which may complete any analysis of temporal resolution below this period. In case you have plans to do this, please mention it in the text. -Otherwise you may ignore this comment.

  **Reply.** We indeed do not plan to add 3-hourly data for scientific reasons. Firstly, because we are primarily interested in precipitation durations of 1-3 hour for benchmarking experiments with convection-permitting simulations. Secondly, with regard to longer rainfall durations, sliding 24-hour accumulations will lead to an underestimation of 24-hour extremes.

- Please employ the alternative method mentioned in l155-157 instead of the one currently used. It is good practice to average additive quantities before taking ratios, in order to avoid undue weight for points with very small numbers in the denominator. Please revise the text accordingly

  **Reply.** We have used the suggested alternative method and arrived at similar conclusions. In the revised version, Equation 1 (page 12) has been changed from

$$\text{Relative intensity change} = \frac{1}{(3\,^\circ\text{C} - 1.5\,^\circ\text{C})}\, \mathbb{E}\left[\frac{z_T^{(3)} - z_T^{(1.5)}}{z_T^{(1.5)}}\right], \tag{1}$$

  to

$$\text{Relative intensity change} = \frac{1}{(3\,^\circ\text{C} - 1.5\,^\circ\text{C})}\, \frac{\mathbb{E}\left[z_T^{(3)} - z_T^{(1.5)}\right]}{\mathbb{E}\left[z_T^{(1.5)}\right]}. \tag{2}$$

- Please mention exactly how you find the 1.5 GWL and 3 GWL periods (probably trivial, but nice for the documentation of methods). Your method for calculating change per degree GWL is a bit unconventional; please discuss how big an effect this has relative to the more frequently used one of taking changes from historical to end-of-century and dividing by whatever global warming happens between those two periods. I

do see advantages in you method related to avoidance of extreme periods in very sensitive simulations, but please add some discussion.

**Reply.** Periods corresponding to a particular GWL are identified as the first 30-year period for which the global average near-surface temperature of the forcing GCM reaches the GWL as compared to the pre-industrial period 1881–1910. The methodology with fixed global warming levels is supposed to be more robust than using fixed time periods as some uncertainty from the climate sensitivity of the GCMs is removed. The explanation of how and why GWL periods are determined can be found in Vautard *et al.* (2014) to which we refer in the revised version.
On Lines 198-200, we added: *"The periods are taken from Vautard* et al. *(2014) and are listed in Table S4. Vautard* et al. *(2014) argue that this methodology with fixed global warming levels is more robust than using fixed time periods as some uncertainty from the climate sensitivity of the GCMs is removed."*

- Please consider using a more diverse colour palette in figs 1 and 2. It is currently very hard to distinguish levels.

  **Reply.** We have defined a more diverse colour palette ourselves As a result, for example, the relatively high values simulated by HadREM3 for the Mediterranean region are much more clearly visible.

**2 Response to Anonymous Referee #2**

**Summary**

First of all thanks to the authors for processing and compiling this useful dataset on annual maximum precipitation amounts for a subset of the EURO-CORDEX RCM ensemble and the comprehensive and extensive accompagnying study, which demonstrates the potential usefulness of the dataset. Despite the fact that the data is basically available through the Earth System Grid Federation data nodes, the data processing and compilation and sharing as FAIR open access research data makes total sense. The manuscript is well written, the dataset is well prepared and fits the scope of the journal. Some open issues as to the construction of the ensemble, the processing of the dataset and the presentation through the data descriptor paper albeit remain.
**Reply.** We thank the reviewer for the overall positive assessment and the relevant comments and suggestions.

**General comments**

The dataset, or rather the data product, is novel and useful to and usable by the community. Based on information provided the data product could be reproduced, if needed; see my comments below, a little bit more detail would be desirable. The data product is presented with enough context to existing literature; with some sections though, more references to existing CORDEX analysis may be useful. The manuscript supports the dataset well with very useful examples.

The dataset quality is fine, the dataset DOI works well, data meet FAIR principles. The data is findable and accessible (after free user registration) through the long-term WDCC storage and dissemination infrastructure, uses compressed netCDF-4 as an interoperable data format, complete with meta data and provenance information as well as version control. Common standards are met. Some notes and recommendations on dataset processing and refinements are given below. The dataset itself is of high quality. There does not seem to be any inconsistency between the paper manuscript and the dataset.

The dataset is useful and usable; again some proposition is made below to increase this further by regridding the data to a common grid. The manuscript is properly structured and clearly written. Methods are described in more detail in a useful appendix. Visual material are OK, some minor comments are given below.
**Reply.** We agree that the dataset could be further improved based on these suggestions, and have therefore generated an extended dataset and requested a new DOI from DKRZ: `doi.org/10.26050/WDCC/EUCOR_prec_v2` (Van de Vyver *et al.*, 2025). The dataset is also available to reviewers via Google Drive and can be accessed anonymously. In particular, we have now supplemented the original dataset with regridded data into a common grid. We have also added evaluation runs (whenever available) and useful variables such as "surface height" (orog) and "land_area_fraction" (sftlf) to all NetCDF files.

**Specific comments**

- P/P2/L44ff: Given the validation and usage examples you provide later, perhaps it makes sense to emphasize that the data is especially useful for extreme value statistics.
  **Reply.** On Line 47, we now refer to extreme value statistics.

- P/P3/L70ff: Perhaps mention that there are many more ensemble members available, but you were after sub-daily data, which are provided by much less ensemble members of the CORDEX-CMIP5 simulations. Maybe also address here, that the CORDEX-CMIP5, other than CORDEX-CMIP6, was not based on some balanced matrix design as described in Katragkou et al. (2024, BAMS), who you cite. So I wonder what motivates eventually the ensemble subset the dataset covers? The EURO-SUPREME dataset does not contain evaluation run results, but especially for using the data for benchmarking, e.g., CPRCM simulations or other validation purposes. Especially for the Section 3 evaluation, this evaluation run dataset might also be helpful. Would it make sense and could this still be added? For completeness also the land-sea masks ("sftlf" variable) and the orography ("orog" variable) data should be provided.
  **Reply.** We provided a more complete description of the selection of the EURO-CORDEX ensemble members.

Firstly, on Lines 72-73, we mention that of the total 69 GCM-RCM combinations in the 0.11° ensemble, only 27 provide precipitation data at an hourly temporal resolution.

Secondly, on Lines 77-79, we explain that a drawback of using a relatively small number of GCM-RCM pairs is that their selection was based solely on data availability, rather than on the principles of the EURO-CORDEX Balanced Ensemble Design (Katragkou *et al.*, 2024), which takes into account additional factors, such as physical plausibility, representativeness of future climate distributions, and the statistical independence of the models.

Thirdly, as mentioned above in our reply to *"General comments"*, we have added on Lines 75-76 evaluation runs, but they are only available for four RCMs: ALADIN63, COSMO-crCLIM, HadREM3-GA7-05 and RCA4. The description of the file names is given on Lines 100-104.

Finally, a land-sea mask ("sftlf" variable) and the orography ("orog" variable) have been added to all NetCDFs and this was mentioned in the text on Lines 107-109.

- P/P3/L75: You mention the data is on different grids, which the reviewer can confirm from personal experience. However, given you want to provide a dataset most easy to use, would it not be better to provide the data ona common grid, e.g., a curvilinear EUR-11 grid, based on the original grid specification of EUR-11 (now EUR-12)? You may regrid using nearest neighbour resampling. This seems to me a weakness of the dataset provided.
  **Reply.** As already mentioned in the *"General comments"*, all simulations were now regridded to a common regular $0.11° \times 0.11°$ grid with spatial coverage $28°N-70°N$ and $13°W-35°E$. This was done using nearest neighbour remapping, using the CDO-command `remapnn`. This has been mentioned on Lines 90-93.

- P/P3/L76ff: Perhaps structure the Section 2.2 a bit more, separating the variables desciption, data formats, data access, etc. from each other. In L92ff your systematic pre-processing / quality checking steps may be described in more detail; right now it reads as if on accident outliers, grid point storms, etc. were detected. Please indicate whether you applied a systematic check.
  **Reply.** Thank you for this useful suggestion. We have restructured and extended the description and divide it into three subsections: Sec. 2.2.1 *"Description of the files and variables"*, Sec. 2.2.2 *"Outliers"*, and Sec. 2.2.3 *"Preliminary visualisation of the dataset"*.
  In particular, in Sec. 2.2.2 we systematically checked the hourly precipitation data for unrealistic values by identifying exceedances above a threshold value of 100 mm/h, that could be attributed to the occurrence of non-physical "grid point storms".

- P/P3/L77: Maybe introduce the block maxima method, later on discussed in the Sect.8.
  **Reply.** This has been introduces on Lines 89-90.

- P/P4/L99: In addition to Fig.1 and Fig.2 perhaps provide an example dataset visualisation, like the 24h maximum annual precipitation for any given year, so the reader gets an idea of the nature of the dataset. Starting off with the return levels, i.e., a drived quantity based on the originally provided data could come after that.
  **Reply.** A specific year of an historical simulation (COSMO-crCLIM) is plotted for hourly maxima in Fig. 1, just before the existing return level maps.

- P/P5/L198ff/Sect.3: Would it also make make sense to cite some of the many studies which have validated and evaluated the EURO-CORDEX CMIP5 RCM ensemble precipitation dataset? In light of the previous Fig.1/2 it makes sense to evaluate the return levels. However, given the seemingly sketchy reference data and the spatial aggregation, would it not make more sense to compare to the very same diagnostic as you provide, annual maximum precipitation per grid element per duration class, and assess typical bias measures? For example for PRUDENCE regions, which you use in Section 5 anyhow? Because given your return level evaluation yields results with larger biases, the reader might question the dataset usefulness. I do think however the dataset is very useful, and it is know that RCMs exhibit biases with precipitation. At the same

time, the return level comparison allows you to compare the statistical properties of the historical run vs observations. So without using an evaluation (ERA-Interim driven) simulation based precipitation diagnostic a direct comparison is difficult as well.

**Reply.** On Lines 139-140, we added references to standard works on the validation of EURO-CORDEX simulations (e.g. Kotlarski et al., 2014; Rajczak and Schär, 2017; Vautard et al., 2021; Barnes et al., 2024).

The advantage of the diagnostics proposed above, which consists of comparing the annual maxima of simulations and observations rather than comparing the corresponding $T$-year return levels, is that it involves less uncertainty, especially for (very) long return periods $T$. However, $T = 10$ years is quite common and acceptable for an evaluation, as previously illustrated in Berg *et al.* (2019) and Poschold *et al.* (2021). Furthermore, the reviewer's suggestion may be a valuable option only if the observed $d$-hourly precipitation extremes are directly available but this is, however, not the case. We must therefore limit ourselves to national statistics.

We also compared the 10-year return levels from the evaluation runs (downscaled from ECMWF-ERAINT) with those from the observations and obtained quantitatively similar results per model in reproducing subdaily rainfall extremes.

- P/P7/L129f: You should make clear you still refer to the return level biases.
  **Reply.** This is changed on Line 161.

- P/P9/L135ff/Sect.4: Would this section perhaps fit better under the Section 5 with application examples as the goal of the paper is primarily on the dataset you present and not the derived climate change analysis?
  **Reply.** Thank you for the suggestion. Sec. 4 *"Example applications"* of the revised manuscript are split into Sec. 4.1 *"Benchmarking CPRCMs"* and Sec. 4.2 *"Changes in EURO-CORDEX rainfall extremes under global warming"*, with the latter section corresponding to Sec. 4 of the original manuscript.
  The purpose of Section 4 is briefly summarized in Lines 168-170.

- P/P12/L171ff: This is an interesting application example. Only I would not consider here the CORDEX-SUPREME the benchmark dataset, it is used in a benchmarking experiment, but the benchmark dataset is the observational reference data from the synop stations.
  **Reply.** Indeed, this may cause confusion. We have therefore replaced "benchmark" with "benchmarking experiment" on Line 173.

- It may not be required by the journal, but as you put a lot of emphasis on the analysis through the return levels, it would be interesting to the reader to also have the code available through a long-term public git repository.
  **Reply.** We extend Sec. 6 *"Code availability"* by specifying which R-packages and functions can be used to fit the GEV-distribution and then calculate the corresponding return levels. Finally, the codes were now made public via the GitHub-repository https://github.com/anodieri/extreme-precipitation-figures.

**Technical corrections**

- P: Out of curiosity, what does "SUPREME" stand for? Does it have a meaning?
  **Reply.** We were not very clear about that. It stands for *"EURO-SUbdaily PRecipitation extrEMEs"* and is being stressed now in the Abstract, Lines 4-5.

- P/P1/L6: replace: "0.11" (redundant with following and information is missing) with "regional climate model (RCM)"
  **Reply.** This is being changed in the Abstract, Line 9.

- P/P1/L6: rephrase: "(coupled to CMIP5)" to "(downscaling CMIP5 GCMs)"
  **Reply.** This is being changed in the Abstract, Line 7.

- P/P1/L6: rephrase: "precipitation depths" to "precipitation amounts"
  **Reply.** This is being changed in the Abstract, Line 7.

- P/P1/L7: rephrase: "More specifically ... by a" to "Specifically, data are based on a 35-member RCM ensemble ..."
  **Reply.** This is being changed in the Abstract, Line 7.

- P/P1/L8: add: "EURO-CORDEX EUR-11 (0.11°) domain"
  **Reply.** This is being changed in the Abstract, Line 9.

- P/P1/L20: missing: reference to last sentence.
  **Reply.** We make a reference to Rajczak and Schär (2017) on Line 21.

- P/P2/L22: remove: "urban water management", just "water management"
  **Reply.** This is being changed in the Introduction, Line 23.

- P/P2/L32: rephrase: call the models CPRCMs, as you refer to RCMs at km-scale resolution
  **Reply.** "CPMs" is changed to "CPRCMs".

- P/P2/L33f: rephrase: the computational resources needed are bigger, this is presumably what you mean; also cite Schaer et al. (2020, https://doi.org/10.1175/BAMS-D-18-0167.1), with an overview of computational demands of CPRCM simulations
  **Reply.** On Line 36, we rephrased this as follows: *"However, the computational resources are greater than that of coarse-resolution RCMs".* We also cite Schär *et al.* (2020).

- P/P2/L34f: add: also the length of the simulations (so far) is rather short, aside from ensemble and domain size
  **Reply.** On Line 38, we add: *"Also, the length of the simulations (so far) is quite short apart from the ensemble and domain size."*

- P/P2/L35ff: there are also other ways to demonstrate the CPRCM added value, I think this is a misleading motivation; the fact is that the CPRCM may simulate extreme precipitation more accurately, albeit they may be of limited use due to the constraints you mention
  **Reply.** We understand that this sentence may be misleading and has little added value. We will therefore remove it.

- P/P2/L39f: mention the CPRCM benchmark objective later on, in line 44f
  **Reply.** This is moved to Lines 168-169.

- P/P2/L44ff: align the listing of purposes with the overview in the abstract, to make the motivation for the dataset very clear
  **Reply.** We aligned the list of purposes in the abstract, on Lines 12-14 and in the Introduction, on Lines 39-41.

- P/P3/L63f: explain the CORDEX acronym, also it is not a "framework" but a WCRP "project", cite Gutowski et al. (2016, https://doi.org/10.5194/gmd-9-4087-2016)
  **Reply.** On lines 65-66, we added: *"...Coordinated Regional Climate Downscaling Experiment (CORDEX) project of the World Climate Research Program (WCRP) (Gutowski Jr. et al., 2016)."*

- P/P3/L84f: It seems you follow the data reference syntax and filename nomenclature as used by CORDEX, which I think is very good. Perhaps you want to indicate this.
  **Reply.** On Line 95, we mentioned the use of the CORDEX nomenclature.

- P/P6,7/Fig1,2: Please use a more differentiating color-scale. Despite adjusting the colorbar range between 1h and 24h annual precipitation maxima return levels, the colormap should be the same for the same kind of variable.
  **Reply.** We redrew Figs. 2 and 3 using a more diverse color palette that we defined ourselves. As a result, for example, the relatively high values simulated by HadREM3 for the Mediterranean region are much more clearly visible.

- P/P6,7/Fig1,2: Please indicate that you do not show the entire EUR-11 CORDEX domain for readers not so familiar with the CORDEX RCM ensemble.
  **Reply.** We indicate in the caption that we show a large part of the EUR-11 CORDEX domain.

- P/P10/Fig.5 caption: return level of what?
  **Reply.** We add to the caption of Fig. 7: *"10-year return levels of hourly and daily precipitation"*

- P: partly there is a mix of British and American English spelling. Please double-check this throughout.
  **Reply.** We double-checked the manuscript and opted for British English.

- P/P12/L190: remove: "over the (southern)" − > "(southern)"
  **Reply.** This is being removed on Line 191.